# Small Fish, Big Answers: Zebrafish and the Molecular Drivers of Metastasis

**DOI:** 10.3390/ijms26030871

**Published:** 2025-01-21

**Authors:** Mayra Fernanda Martínez-López, José Francisco López-Gil

**Affiliations:** 1School of Medicine, Universidad Espíritu Santo, Samborondón 092301, Ecuador; 2One Health Research Group, Universidad de Las Américas, Quito 170124, Ecuador

**Keywords:** zebrafish, cancer, metastasis, tumor microenvironment, angiogenesis, personalized medicine

## Abstract

Cancer metastasis is a leading cause of cancer-related deaths and represents one of the most challenging processes to study due to its complexity and dynamic nature. Zebrafish (*Danio rerio*) have become an invaluable model in metastasis research, offering unique advantages such as optical transparency, rapid development, and the ability to visualize tumor interactions with the microenvironment in real time. This review explores how zebrafish models have elucidated the critical steps of metastasis, including tumor invasion, vascular remodeling, and immune evasion, while also serving as platforms for drug testing and personalized medicine. Advances such as patient-derived xenografts and innovative genetic tools have further established zebrafish as a cornerstone in cancer research, particularly in understanding the molecular drivers of metastasis and identifying therapeutic targets. By bridging the experimental findings with clinical relevance, zebrafish continue transforming our understanding of cancer biology and therapy.

## 1. Introduction

Cancer metastasis, the spread of tumor cells from a primary site to distant organs, is a multistep and highly dynamic process that is responsible for the majority of cancer-related deaths [1,2,3]. The metastatic process occurs through a series of defined stages (Figure 1): cancer cells invade the surrounding tissues, enter the bloodstream or lymphatic system (intravasation), survive in the circulation, exit into distant tissues (extravasation), and establish secondary tumors in new tissues (colonization) [1]. Throughout these stages, tumor cells must overcome mechanical, biochemical, and immune challenges while interacting with their surrounding microenvironment [1]. Gaining a deeper insight into these complex processes is essential for developing strategies to prevent and control metastasis.

The tumor microenvironment (TME) is central to metastatic progression, consisting of a complex network of stromal cells, an extracellular matrix (ECM), immune cells, and soluble signaling molecules [4]. These components can either inhibit or support metastasis, depending on the specific context, which adds to the challenge of fully understanding this process and developing effective treatments [5].

Despite significant progress in understanding metastatic processes, many aspects remain poorly understood, largely due to the challenges of studying each step in vivo [6]. Traditional models like genetically engineered mouse models (GEMMs) and patient-derived xenografts (PDXs) have been valuable in exploring metastasis biology and testing anti-cancer therapies [7]. However, these models have inherent limitations, including high costs, lengthy experimental timelines, and real-time challenges in capturing the dynamic interactions between tumors and the TME [8]. Recent advances have improved the resolution and temporal analysis of these models. Still, these improvements often require invasive surgical procedures and specialized imaging technologies, which can hinder the reproducibility of the studies [9].

Zebrafish (*Danio rerio*) have emerged as a powerful and versatile model system in cancer research, addressing the numerous limitations associated with traditional models [10]. As a small vertebrate, the zebrafish share highly conserved biological features with humans, including organ systems such as the central nervous system [11], intestines, pancreas, and liver [12]; metabolic pathways like glucose [13] and lipid [14] homeostasis; and physiological functions, including cardiac electrophysiology and heart rate regulation [15]. Additionally, this model presents a high conservation of key human cancer-related genes, such as *tumor protein p53* (*TP53*) [16], *Kirsten rat sarcoma viral oncogene homolog* (*KRAS*) [17], and *vascular endothelial growth factor-A* (*VEGFA*) [18], and critical signaling pathways, including phosphatidylinositol 3-kinase (PI3K)/protein kinase B (AKT) (proliferation) [19], RHOA/Rho-associated protein kinase (ROCK) (migration) [20], and B-cell leukemia/lymphoma 2 protein (BCL2) (cell death) [21]. The pathways involved in differentiation (e.g., neurogenic locus notch homolog protein–NOTCH [22]) and immune responses, such as danger-associated molecular pattern (DAMP) signaling [23], are also preserved. These similarities make zebrafish particularly valuable in studying cancer progression and metastasis. One of their most notable advantages is the optical transparency of embryos, which enables the real-time, single-cell high-resolution imaging of critical metastatic events, such as tumor cell invasion, intravasation, extravasation, and migration [10] (Figure 2). Their rapid external embryonic development and high fecundity facilitate large-scale experimental designs and high-throughput approaches, allowing researchers to test hypotheses and identify the mechanisms involved in tumor biology efficiently [10,24]. Zebrafish are also highly amenable to genetic manipulation, with tools like CRISPR/Cas9 enabling the modeling of specific genetic alterations observed in human cancers [25].

Zebrafish models have been widely utilized to study metastatic cancers, offering a valuable platform to investigate tumor dissemination and organotropism [26,27]. In breast cancer, they have elucidated the mechanisms of brain and bone metastasis [28]. Prostate cancer models have revealed conserved genomic alterations in the metastatic site [29], and melanoma studies have explored drivers of tumor invasion [30]. Zebrafish have also provided insights into rarer cancers like glioblastoma and pediatric tumors [31,32,33], highlighting their real-time imaging capabilities to comprehensibly study metastatic behavior.

In addition to their relevance to metastatic cancer, zebrafish support the engraftment of human tumor cells [34,35] and retain the key components of the TME, such as fibroblasts, macrophages, and neutrophils [36]. This allows for the study of tumor–stroma interactions in a dynamic and physiologically relevant setting. Furthermore, zebrafish replicate specific aspects of human cancer biology, including organ-specific metastasis, such as liver and brain tropism [37], making them an invaluable tool for understanding the mechanisms underlying metastatic spread and developing novel therapeutic strategies. Their cost-effectiveness and accessibility [10] further enhance their appeal, enabling resource-limited laboratories to contribute to cutting-edge cancer research without compromising on experimental rigor.

This review examines how zebrafish have transformed metastasis research by offering unique insights into tumor–TME interactions, enabling advanced imaging and genomic studies, and serving as an efficient platform for high-throughput drug testing. It also highlights the model’s strengths and limitations, underscoring its vital role in connecting preclinical research with clinical applications.

**Figure 1 ijms-26-00871-f001:**
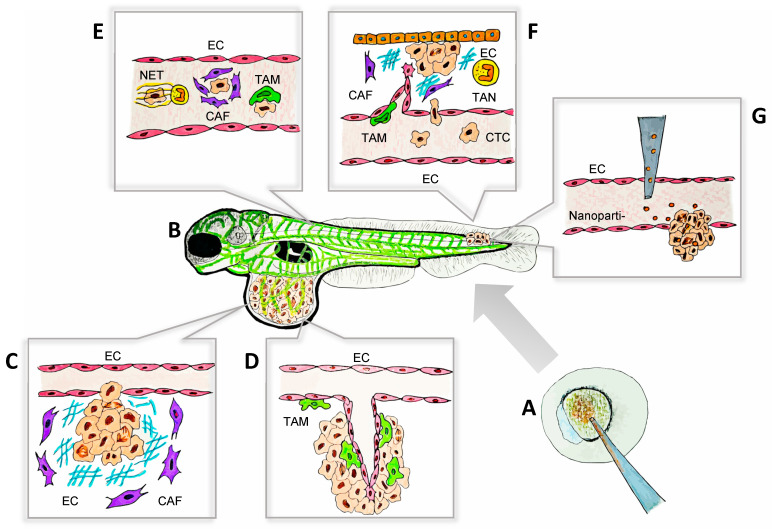
**The zebrafish model as a platform for the visualization of tumor–TME interactions during metastasis.** The key aspects of tumor–TME interactions, including ECM remodeling, angiogenesis, immune cell recruitment, and tumor invasion, can be visualized in real time and at a single-cell resolution using zebrafish models. This approach facilitates high-throughput drug testing and provides a dynamic, cost-effective platform for studying metastasis. The processes illustrated in this schematic are examples of observable cell behaviors studied in zebrafish models that have deepened our understanding of metastasis and advanced personalized therapies. (**A**) Mutant and transgenic zebrafish can be generated with ease due to the accessibility of embryos at the single-cell stage. Techniques such as morpholino oligonucleotides, CRISPR/Cas9, and other gene-editing tools are readily applicable in this model. (**B**) Schematic representation of a transgenic larval zebrafish xenograft model, where blood vessels are labeled with a fluorescent green protein, enabling clear visualization of the primary tumors and metastatic sites. This set-up facilitates the real-time monitoring of tumor progression and metastasis with a high resolution. (**C**) CAFs (purple) remodeling the ECM in the primary tumor to facilitate tumor extravasation. (**D**) TAMs (green) interact closely with ECs, promoting angiogenesis in the primary tumor through direct contact and the release of soluble factors, including VEGF-A. (**E**) CTCs survive their journey to the metastatic sites with the support of NETs (yellow), CAFs (purple), and TAMs (green). These components create a protective microenvironment, shielding CTCs from immune system attacks and other potential threats, thereby facilitating their successful dissemination and colonization. (**F**) CTCs extravasate and colonize distant tissues with the support of soluble factors secreted by TAMs (green), TANs (yellow), and CAFs (purple). Furthermore, TAM-induced angiogenesis facilitates tissue invasion and supports the establishment of the metastatic niche, promoting successful tumor colonization. (**G**) Nanoparticle-based drug delivery in zebrafish models, targeting tumor cells in a metastatic site for therapeutic evaluation. CAF: cancer-associated fibroblast; CTC: circulating tumor cell; EC: endothelial cell; ECM: extracellular matrix; NET: neutrophil extracellular trap; TAM: tumor-associated macrophage; TAN: tumor-associated neutrophil; TME: tumor microenvironment.

## 2. Advanced Molecular and Cellular Insights of Metastasis: From Mechanisms to Modeling

Metastasis is a highly regulated process influenced not only by the intrinsic properties of tumor cells but also by their interactions with the surrounding cells and molecules, many of them being part of the TME [4]. Importantly, the components of the TME greatly contribute to the success or failure of metastatic dissemination [4,38]. Additionally, tumor–TME interactions are mediated by a range of molecular players, including cytokines, growth factors, proteases, and adhesion molecules, which regulate crucial processes such as ECM remodeling, angiogenesis, immune evasion, and vascular invasion [38,39]. Zebrafish models have proven to be essential tools for studying these mechanisms in vivo, providing key insights into the molecular dynamics of this crosstalk during metastasis [36] (Table 1; Figure 1 and Figure 2). Moreover, the integration of cutting-edge technologies into zebrafish models has significantly expanded their utility in cancer research [40]. Zebrafish now serve as versatile platforms for genomic and proteomic analyses, the real-time imaging of tumor–TME interactions, and high-throughput drug discovery pipelines (Figure 1) [41]. Using tools such as CRISPR screens, RNA sequencing (RNA-seq), and spatial transcriptomics, along with PDX models and advanced drug delivery systems, zebrafish enable groundbreaking insights into metastasis that were previously not easily accessible.

### 2.1. Exploring Tumor Invasion Through the Modeling of ECM Dynamics

Cancer-associated fibroblasts (CAFs) play a significant role in metastasis by altering the composition of the ECM and secreting signals that enhance tumor cell invasiveness (Figure 1C) [42]. Through the production of matrix metalloproteinases (MMPs), such as MMP2 and MMP9, CAFs break down ECM barriers, facilitating tumor cell migration and invasion [42,43]. In addition to structural changes, CAFs release cytokines, including transforming growth factor beta (TGF-β) and interleukin (IL)-6, which promote epithelial-to-mesenchymal transition (EMT), a critical process in increasing tumor cell motility [44]. The role of MMPs in metastasis was further corroborated by exploring cooperative invasion in melanoma [45]. When co-grafted into zebrafish with highly invasive melanoma cells, poorly invasive melanoma cells acquired invasive properties. This process relied on MMP activity and tumor-derived ECM factors, such as fibronectin and integrin [45]. These findings highlight the critical role of CAFs and MMP activity in remodeling the TME, driving cancer cell invasiveness and facilitating metastatic progression. Furthermore, CAFs enhance circulating tumor cells’ (CTCs) survival and proliferation by producing pro-survival factors such as regulated upon activation, normal T cell expressed and secreted (RANTES), and intercellular adhesion molecule 1 (ICAM1) [46]. Heterotypic CTC–CAF clusters support tumor cell survival (Figure 1C) and proliferation more effectively than homotypic clusters, emphasizing the role of CAF-mediated stromal contributions in metastasis [46].

Zebrafish models have been instrumental in uncovering these processes, offering unique advantages for studying ECM properties, stromal stiffness, and tumor cell motility. Advanced loss-of-function techniques, such as morpholino oligonucleotides (MOs) and short-hairpin RNA (shRNA), have proven particularly valuable in this context [47]. MOs enable the transient suppression of gene expression [48], allowing researchers to investigate the genes involved in ECM remodeling and angiogenesis. Meanwhile, shRNA-mediated knockdown provides a longer-term solution for gene silencing, making it possible to study the effects of specific genes on tumor progression and metastasis [49]. For example, in zebrafish xenograft models of oral squamous cell carcinoma, shRNA-mediated silencing of MMP9 led to reduced ECM degradation, tumor invasion, and metastatic spread. This intervention significantly decreased the number of metastatic lesions, further reinforcing the essential role of MMP9 in metastasis [50]. Together, these approaches demonstrate the power of zebrafish as a model for studying CAF activity, ECM dynamics, and therapeutic interventions targeting the metastatic cascade.

### 2.2. Real-Time Insights into Angiogenesis and Vascular Dissemination

Angiogenesis, forming new blood vessels from the existing vasculature, is a critical step in cancer progression. It sustains tumor growth by supplying essential nutrients and oxygen while facilitating the entry of tumor cells into the circulation [51]. In zebrafish xenografts, tumors initially develop as avascular masses until they reach a critical size, approximately 100 cells, at which point they recruit endothelial cells to form new blood vessels [52]. This process, known as the angiogenic switch, supports rapid tumor growth. Interestingly, smaller tumors with invasive behavior can bypass angiogenesis altogether, enabling cancer cells to extravasate and form micrometastases in distant tissues [52]. These findings emphasize how the tumor size and cell number influence metastasis, particularly the reliance on angiogenesis for tumor progression.

Zebrafish models offer unparalleled opportunities to observe and manipulate angiogenesis and tumor progression in real time. Among the various genetic tools, the Gal4-UAS system has emerged as a precise approach for controlling gene expression. By utilizing the Gal4 transcription factor to activate UAS-linked genes, this system enables targeted expression in specific tissues or at defined developmental stages [53]. When paired with transgenic reporter lines, the Gal4-UAS system allows for the real-time visualization of signaling pathways, such as TGFβ and Notch, which play critical roles in angiogenesis, tumor progression, and metastatic dissemination [54].

The use of stable transgenic lines facilitates long-term studies of dynamic tumor-microenvironment interactions, including stromal recruitment, immune modulation, and vascular remodeling [55]. Techniques such as the Tol2 transposon system and CRISPR/Cas9-mediated insertion [25,53,56] enhance the generation of zebrafish lines with specific oncogene expression or fluorescently labeled cell populations, such as endothelial cells (Figure 2A–C), further advancing the study of tumor–TME interactions at a single-cell resolution in real time [55]

#### 2.2.1. Key Molecular Drivers of Angiogenesis

##### The VEGF Pathway

The VEGF family is a major driver of tumor-induced angiogenesis. VEGF-A, the most studied isoform, binds to its primary receptor VEGFR-2 on endothelial cells, activating signaling pathways such as PI3K/AKT and mitogen-activated protein kinase (MAPK). These pathways promote endothelial cell proliferation, migration, and survival [57]. VEGF-A also increases vascular permeability, enabling tumor cells to extravasate into the circulation [58,59]. Hypoxia within tumors upregulates VEGF-A via the stabilization of hypoxia-inducible factor 1-alpha (HIF-1α), creating a feedback loop that sustains angiogenesis [60,61,62]. VEGF overexpression is a hallmark of numerous malignancies. In hematological cancers, such as leukemia, lymphoma, and multiple myeloma, VEGF functions as a key autocrine and paracrine factor driving tumor growth and angiogenesis [63]. Its overexpression correlates with therapeutic resistance and poor prognosis [63,64,65]. Similarly, in solid tumors, including gastrointestinal adenocarcinomas and cancers of the lung, prostate, and ovary, VEGF overexpression is associated with aggressive disease progression and unfavorable outcomes [66,67,68,69].

Functional studies using zebrafish models have reinforced the critical role of VEGF in vascular remodeling, an essential step in the metastatic cascade. For instance, VEGF-A MO silencing in zebrafish xenografts of breast cancer and melanoma revealed the involvement of macrophages in promoting angiogenesis-driven tumor progression, highlighting the interplay between immune cells and tumor-derived signals [70,71]. Moreover, VEGFR-2 knockdown using shRNA in zebrafish models significantly disrupted vessel formation, underscoring the critical role of this signaling pathway in angiogenesis [72].

Interestingly, while VEGF inhibition can suppress angiogenesis, it may paradoxically enhance invasiveness. For example, VEGFR2/3 inhibition in breast cancer xenografts reduced lymph node metastases by decreasing the lymphatic vasculature but simultaneously increased metastases in other organs, such as the diaphragm and pancreas [73]. Similar compensatory upregulation of alternative pro-angiogenic pathways has been observed in mouse models of pancreatic neuroendocrine carcinoma and glioblastoma (GBM) following VEGF inhibition, likely driven by hypoxia-induced stress [74]. VEGF also disrupts the VE–cadherin–β-catenin complex, further facilitating tumor cell extravasation and metastasis [75].

##### Hypoxia

Due to their inherently aberrant structural characteristics, most solid tumors exhibit areas of significant hypoxia, one of the hallmarks of cancer, particularly in the tumor’s interior regions farthest from the potential blood supply [76]. Hypoxia plays a dual role in driving angiogenesis and metastasis. While hypoxic conditions initially cause vessel regression, they can also stimulate angiogenesis and enhance tumor invasion [77]. In zebrafish, tumors exposed to hypoxic conditions for three days become highly angiogenic, initiating extravasation, invasion, and metastasis, like VEGF-inhibition-induced metastasis [77,78]. Hypoxia-driven changes in the TME, including oxidative stress and VEGF upregulation, can promote vascular regrowth and further metastasis following anti-angiogenic therapy [79]. Additionally, under hypoxia, cancer cells shift from oxidative phosphorylation to glycolysis, a phenomenon known as the Warburg effect, leading to increased glucose uptake and lactate production [80,81]. This metabolic shift not only fuels tumor growth but also creates an acidic extracellular environment that promotes ECM degradation, angiogenesis, and immune evasion [80,81]. Moreover, hypoxia stabilizes HIF-1α, which upregulates the genes involved in glycolysis, glutaminolysis, and lipid biosynthesis [82,83,84]. These changes enhance the ability of tumor cells to adapt to nutrient deprivation and oxidative stress, thereby facilitating their invasion and metastasis [80].

Zebrafish models uniquely enable the study of hypoxia-induced metabolic shifts, and their influence on tumor progression and metastasis. These models facilitate the real-time analysis of glycolysis, oxidative phosphorylation, and lipid metabolism in vivo, revealing how metabolic reprogramming supports angiogenesis, ECM remodeling, and immune evasion. This approach provides critical insights for identifying precise therapeutic targets to inhibit metastasis at its metabolic roots.

##### Angiopoietin and the TIE Pathway

Angiopoietin-2 (ANGPT2) destabilizes vascular networks by loosening endothelial junctions, preparing vessels for remodeling and sprouting [85]. ANGPT2 interacts with the TEK tyrosine kinase (TIE) 2 receptor, amplifying VEGF-driven angiogenesis. Elevated serum concentrations of ANGPT2 have been observed in patients with multiple myeloma [86], non-small cell lung cancer (NSCLC) [87], metastatic colorectal cancer (CRC) [88], renal cell carcinoma [89], hepatocellular carcinoma [90], and chronic lymphocytic leukemia [91], all of which are associated with a poorer prognosis and rapid disease progression. On the other hand, anti-angiogenic factors like thrombospondin-1 (TSP-1) and endostatin are often downregulated in tumors, tipping the balance toward vascular growth [92,93]. High-resolution imaging of zebrafish with fluorescently labeled endothelial cells showed how ANGPT2 destabilizes vessels and enhances VEGF-mediated sprouting [94]. These findings highlight the critical role of ANGPT2 in vascular destabilization and its contribution to forming abnormal, leaky networks that support tumor growth and metastasis.

##### ECM Remodeling and MMPs

The MMP-mediated degradation of the basement membrane and ECM is a prerequisite for angiogenesis, clearing pathways for migrating endothelial cells [27]. MMP expression is also directly correlated with a higher metastatic potential in several cancers including CRC, NSCLC, breast cancer, and prostate cancer [95]. Stromal cells, including CAFs and tumor-associated macrophages (TAMs), secrete MMP2 and MMP9, driving ECM degradation [27,96,97,98]. As previously described, the inhibition of MMP activity reduces vascular remodeling and impairs angiogenesis, further confirming the role of ECM dynamics in endothelial migration [55].

Furthermore, zebrafish with fluorescent vasculature reporters (Figure 1B and Figure 2B,C) are essential for studying candidate genes in angiogenesis, showing how tumor and TME signals shape leaky vascular networks that drive metastasis [99]. These findings underscore the complex mechanisms of angiogenesis and its role in metastatic progression.

#### 2.2.2. Endothelial–Tumor Interactions

The interplay between tumor cells and endothelial cells is central to metastasis. Tumor cells adhere to endothelial surfaces through adhesion molecules such as E-selectin, ICAM-1, and vascular cell adhesion molecule 1 (VCAM-1), facilitating both intravasation and extravasation [100]. Tumor-derived extracellular vesicles (EVs) carry pro-angiogenic factors, such as VEGF and ECM remodeling enzymes, that modulate endothelial behavior and promote vascular permeability [101,102].

Zebrafish provide a precise platform for studying endothelial–tumor interactions, with EMT standing out as a critical process in these dynamics [103,104]. This transition, mediated by TGF-β signaling, enhances endothelial plasticity and contributes to the invasive front [105,106]. Knockdown experiments targeting TGF-β in zebrafish revealed reduced metastasis due to less EMT [106]. Additionally, timelapse imaging has captured how VEGF gradients direct vascular sprouting, with tip endothelial cells exhibiting high VEGFR-2 expression and responding to stromal-derived factors like fibroblast growth factors (FGFs) and angiopoietins [107].

The centrality of angiogenesis in tumor progression has made it a prominent target for cancer therapies. Inhibitors of VEGF, such as bevacizumab [108], and small-molecule tyrosine kinase inhibitors targeting VEGFR, such as sunitinib and sorafenib [109], have demonstrated efficacy in reducing tumor vascular density and delaying progression. However, these drugs are often associated with significant side effects and toxicities in clinical settings, including hypertension, proteinuria, bleeding, and impaired wound healing [110,111,112]. Resistance mechanisms further complicate treatment, as tumors can upregulate alternative angiogenic mediators, such as ANGPT2 or basic FGF, to bypass VEGF inhibition [112].

Zebrafish models have been instrumental in advancing research to address these challenges. These models allow the study of compensatory angiogenic pathways and enable the high-throughput screening of drug combinations that could mitigate side effects [34,110,113]. For example, zebrafish studies have shown that dual inhibition of VEGF and ANGPT2 significantly reduces vascular density compared with single-agent therapies, offering potential strategies to overcome resistance while minimizing toxicity [114]. Additionally, zebrafish xenograft models were used to screen combinations of VEGF inhibitors with MMP blockers, providing preclinical evidence for the synergistic anti-angiogenic effects [111,115]. Furthermore, zebrafish xenografts demonstrated high potential as efficient screening platforms for anti-angiogenic therapies like bevacizumab [110]. These models enabled the rapid evaluation of tumor angiogenesis, micrometastasis, and therapeutic responses, while also offering insights into patient-specific outcomes, making them highly valuable for developing targeted therapies with fewer side effects for personalized medicine [110].

Advances in genome editing, particularly CRISPR/Cas9 technology, have further enhanced the utility of zebrafish models in metastasis research [116]. The Cas9 nuclease, guided by a single-guide RNA (sgRNA), creates targeted double-stranded DNA breaks. These breaks are repaired by either non-homologous end joining (NHEJ), which often results in gene knockouts through insertions or deletions, or homology-directed repair (HDR), enabling precise edits like specific mutations or reporter tags [117]. In zebrafish, CRISPR/Cas9 is delivered as either RNA or pre-assembled ribonucleoprotein (RNP) complexes into one-cell-stage embryos, ensuring a high efficiency of mutagenesis in developing tissues [25]. This approach allows for the rapid generation of knockouts for candidate genes involved in metastasis, such as those associated with angiogenesis, ECM remodeling, and chemotaxis [25].

In addition, zebrafish models are particularly valuable for studying metastasis due to their capacity for high-throughput analysis [113]. When combined with CRISPR technology, this feature could enable the rapid, parallel screening of multiple genes and pathways, potentially providing deeper insights into their roles in metastatic progression [118]. Advances in genome editing, such as RNP complexes for precise CRISPR mutagenesis, can enhance this model by ensuring high mutagenesis rates with minimal off-target effects, enabling the efficient generation of functional mutants [116]. This is particularly advantageous in metastasis research, where rapid results are needed to link molecular changes to functional outcomes.

### 2.3. Immune Cells: Dynamic Regulators of Tumor Progression

Immune cells perform diverse roles that evolve with tumor progression. Initially, many immune cells exhibit anti-tumor activity, but molecular signals from the TME often co-opt these cells to support tumor growth and metastasis [119]. TAMs and neutrophils are particularly notable for their dual roles, toggling between tumor suppression and promotion based on cues from the TME [120,121].

An important breakthrough in studying immune subsets and their interactions within the TME is the advent of single-cell RNA sequencing (scRNA-seq). This technique allows for the precise profiling of tumor, stromal, vascular, and immune cells at a single-cell resolution, uncovering individual transcriptional states and signaling pathways [122,123]. By offering an unprecedented level of detail, scRNA-seq has significantly deepened the understanding of the complex immune-related mechanisms underlying tumor progression and metastasis. For example, scRNA-seq analysis of GBM zebrafish xenografts revealed significant differences in tumor cell transcriptional states across patient samples. These findings highlighted the specific molecular changes linked to tumor progression, such as increased expression of immunosuppressive genes *galectin 1* (*lgals1*), *triggering receptor expressed on myeloid cells 2* (*trem2*), as well as pathways related to extracellular ECM remodeling. Moreover, zebrafish models of CRC have shown how immune interactions within the TME can reshape the genomic profile of immunogenic subclones into a non-immunogenic state, enabling the evasion of immunosurveillance [124].

#### 2.3.1. Macrophages: Orchestrators of Metastatic Spread

Macrophages are among the most versatile immune cells in the TME, responding dynamically to diverse signals [120] (Figure 2D). In metastatic contexts, macrophages are often polarized into a M2-like phenotype, characterized by the secretion of IL-10, TGF-β, and colony stimulating factor 1 (CSF-1), which suppress inflammation and enhance tumor progression [125]. One way that TAMs promote metastasis is by releasing VEGF, which stimulates angiogenesis and, in turn, facilitates tumor cell migration (Figure 1D) [55,70,77]. Additionally, TAMs physically aid tumor cell intravasation, as in vivo imaging of zebrafish cancer models showed macrophages guiding tumor cells to blood vessels and disrupting the endothelial barriers for vascular entry (Figure 1E,F) [70,125]. These interactions were driven by the C-X-C chemokine receptor type 4 (CXCR4)/C-C motif ligand 2 (CCL2) axis and facilitated by ECM degradation through TAM-secreted MMPs, with MMP9 playing a key role [96,126].

In distant organs, TAMs are essential for pre-metastatic niche formation (Figure 1F). Tumor-derived signals, including CCL2 and TGF-β, recruit macrophages to future metastatic sites, remodeling the stroma and suppressing local immune responses [33,127,128]. Live imaging revealed that microglia and macrophages actively contribute to GBM progression by migrating to tumor sites and closely interacting with cancer cells. Through non-phagocytic interactions, these immune cells promote tumor cell survival and enhance their invasiveness [129]. Besides, TAMs deposit ECM proteins, like fibronectin 1, which promote tumor colonization and metastasis [130]. Additionally, melanoma xenograft models demonstrate that tumor-derived EVs deliver metastasis-promoting proteins to macrophages and endothelial cells, further driving the metastatic cycle [101].

Finally, scRNA-seq of TAMs in GBM models showed that they exhibited diverse immunosuppressive gene profiles including the differential expression of *lgals1*, which supports tumor survival, immune evasion, and cancer progression [122]. These findings emphasize the cellular heterogeneity within the TME and underscore its critical role in determining metastatic behavior and immune escape.

#### 2.3.2. Neutrophils: Partners in Tumor Dissemination

Neutrophils play a crucial role in metastasis, responding to chemokine gradients like CXCL8 (IL-8) and stromal cell-derived factor 1 (CXCL12) to infiltrate the TME [121] (Figure 2E). Although they can exhibit anti-tumor activity in certain conditions, neutrophils are often reprogrammed to support metastasis [121]. A key feature of this pro-tumoral behavior is the formation of neutrophil extracellular traps (NETs), composed of DNA, histones, and proteases such as elastase and MMP9 [131]. These NETs capture CTCs, promote their adhesion to endothelial surfaces, shield them from immune clearance, and, as shown by the real-time tracking of fluorescently labeled neutrophils and tumor cells, guide CTCs to blood vessel walls to prepare them for extravasation (Figure 1E) [132].

Additional observations in zebrafish xenografts revealed that neutrophils are recruited to specific sites, such as the caudal hematopoietic tissue (CHT), the equivalent of the fetal liver in mammals, where they facilitate vascular remodeling and enhance metastatic spread efficiency (Figure 1F) [133]. The CXCL12/CXCR4 axis plays a crucial role in neutrophil-driven metastasis. Tumor cells frequently overexpress CXCL12, which binds to CXCR4 on neutrophils, directing their recruitment and facilitating interactions with tumor cells [134]. Blocking CXCR4 signaling disrupts these interactions, effectively halting early metastatic events [135].

Building on these findings, spatial transcriptomics provides a valuable approach to exploring how these interactions are organized within the TME [136]. This technique has the potential to enable the precise mapping of gene expression and chemokine gradients such as CXCL12 in the vicinity of tumor cells and the surrounding stromal components in zebrafish models [137]. By examining how CXCL12 gradients direct neutrophil and tumor cell migration in vivo, deeper insights can be gained into the spatial dynamics of metastasis [138]. Furthermore, spatial profiling allows for the simultaneous analysis of tumor–stroma interactions, identifying stromal cell populations such as endothelial cells and fibroblasts that contribute to vascular remodeling, immunosuppression, and the establishment of the metastatic niche [139]. This comprehensive approach not only has the potential to reveal the molecular drivers of invasion and dissemination but also to highlight how neutrophil recruitment and chemokine-driven signaling coordinate to facilitate metastatic progression [139].

Moreover, in zebrafish models of HRas^V12^-transformed epithelial cells, neutrophils promoted EMT by increasing the expression of genes and proteins such as *slug*, *vimentin*, and MMP9. This process was triggered by IL-8, which is upregulated during oncogenic transformation and recruits neutrophils to the TME, initiating tumor cell dissemination [140].

**Table 1 ijms-26-00871-t001:** Stages of metastasis: mechanisms and therapeutic insights.

Stage of Metastasis	Description	Key Molecules/Processes	Zebrafish Model Contributions	Therapeutic Implications	Key Differences with Humans	References
**Invasion**	Tumor cells breach the ECM and invade the surrounding tissues	MMPs (MMP2, MMP9), integrins, fibronectin, TGF-β, IL-6, EMT	CAFs and TAMs promote ECM degradation through MMP2/9. IL-8-mediated recruitment of neutrophils promotes EMT and tumor dissemination. Cooperative melanoma invasion involves ECM factors like fibronectin and integrins.	Development of MMP inhibitors to reduce ECM breakdown. Targeting of integrin interactions to limit adhesion and migration. Potential selective inhibition of neutrophil migration towards the TME.	The genomic diversity of MMPs in humans is higher than in zebrafish. TGF-β is mostly found in scar formation processes in humans whereas in zebrafish it is associated with tissue regeneration. Zebrafish have a high regenerative capacity associated with EMT processes, which has not been described in humans.	[45,50,77,78,98,101,102,105,106,129,135,140,141,142,143]
**Intravasation**	Tumor cells enter the blood or lymphatic vessels	VEGF/VEGFR, ICAM-1, VCAM-1, E-selectin, CXCL12/CXCR4, EVs	Tumor-derived EVs carrying VEGF and remodeling enzymes promote endothelial permeability. TAMs stimulate angiogenesis and guide tumor cells to blood vessels, disrupting endothelial barriers for intravasation. CXCL12/CXCR4 signaling recruits neutrophils to tumor cells, facilitating early metastatic events.	Anti-VEGF therapies to reduce vascular permeability. Blocking CXCL12/CXCR4 to disrupt tumor–endothelial interactions. Inhibition of EV release/trafficking.	Zebrafish have two VEGF paralogs (vegfa and vegfb) and humans have three (VEGF-A, VEGF-B, and VEGF-C). Zebrafish have four genes related to VEGFR (flt1, kdr, flt4 and kdrl) and humans have two (VEGFR-1 and VEGFR-2).	[70,101,102,125,135,144,145]
**Circulation**	Tumor cells survive as CTCs	RANTES, IL-6, IL-8, NETs, TGF-β	TAMs guide tumor cells to blood vessels, disrupting the endothelial barriers for intravasation. TAMs and microglia promote tumor invasiveness through non-phagocytic interactions. NETs capture CTCs and promote adhesion, immune evasion, and extravasation. CAFs enhance CTC survival via RANTES and ICAM1.	Targeting of pro-survival factors to reduce CTC viability. NET inhibitors. Immunomodulators to polarize TAMs towards an M1-like phenotype.	TGF-β is mostly found in scar formation processes in humans whereas in zebrafish it is associated with tissue regeneration.	[46,70,125,129,132,143]
**Extravasation**	Tumor cells exit blood vessels to colonize tissues	VEGF/VEGFR, ANGPT2/TIE2, MMP2, MMP9, EVs.	VEGF-induced vascular leakiness promotes tumor cell extravasation. Hypoxia-triggered angiogenesis drives extravasation. ANGPT2 destabilizes vascular networks by loosening endothelial junctions, facilitating vessel remodeling. CAFs and TAMs induce ECM degradation via MMP2/9 and VEGF, enhancing angiogenesis and tumor cell migration.	Combined VEGF/ANGPT2 inhibitors to stabilize vessels. Inhibition of EV release/trafficking. Development of MMP inhibitors to reduce ECM breakdown.	Zebrafish have two VEGF paralogs (vegfa and vegfb) and humans have three (VEGF-A, VEGF-B, and VEGF-C). Zebrafish have four genes related to VEGFR (flt1, kdr, flt4, and kdrl) and humans have two (VEGFR-1 and VEGFR-2). Ang1, Ang2a, and Ang2b are ligands for Tie1 and 2 in zebrafish. In humans, the ligand for TIE2 is ANG4. The genomic diversity of MMPs in humans is higher than in zebrafish.	[27,70,77,78,85,94,98,141,144,145]
**Colonization**	Tumor cells home distant tissues and grow into secondary tumors	CXCL12/CXCR4, IL-10, TGF-β, CSF-1, VEGF, MMPs	Pre-metastatic niche formation by promoting ECM degradation and angiogenesis. TAMs allow for cancer cell homing of the new tissue by inducing immunosuppression through a M2-like phenotype.	Potential selective inhibition of macrophage and/or neutrophil migration towards the TME. Inhibition of TGF-β. Development of MMP inhibitors to reduce ECM breakdown.	TGF-β is mostly found in scar formation processes in humans whereas in zebrafish it is associated with tissue regeneration. The genomic diversity of MMPs in humans is higher than in zebrafish.	[27,50,73,94,98,106,107,125,141,143]
**Angiogenesis**	New blood vessels form from existing ones to irrigate tumors	VEGF/VEGFR2, HIF-1α, ANGPT2/TIE2, TSP-1	VEGF-induced vessel formation and hypoxia-driven angiogenesis. The angiogenic switch, triggered at ~100 cells, promotes rapid growth, while smaller tumors bypass angiogenesis, forming micrometastases. ANGPT2 destabilizes vessels, enhancing VEGF sprouting. VEGF gradients direct vascular sprouting via VEGFR-2 on tip endothelial cells. Neutrophils recruited to the caudal hematopoietic tissue facilitate vascular remodeling.	Platform for high-throughput testing of anti-angiogenic therapies (e.g., bevacizumab, VEGFR inhibitors). Dual targeting of VEGF and ANGPT2 pathways to overcome resistance mechanisms.	Zebrafish have four genes related to VEGFR (flt1, kdr, flt4, and kdrl) and humans have two (VEGFR-1 and VEGFR-2). Ang1, Ang2a, and Ang2b are ligands for Tie1 and 2 in zebrafish. In humans, the ligand for TIE2 is ANG4.	[52,70,72,73,77,78,85,94,101,102,107,110,111,115,133,144,145]

ANGPT2: Angiopoietin-2; CAF: Cancer-associated fibroblast; CSF-1: Colony-stimulating factor 1; CTC: Circulating tumor cell; CXCL12: C-X-C motif chemokine ligand 12; CXCR4: C-X-C chemokine receptor type 4; ECM: Extracellular matrix; EMT: Endothelial-to-mesenchymal transition; EV: Extracellular vesicle; flt: FMS-like tyrosine kinase; HIF-1α: Hypoxia-inducible factor 1-alpha; ICAM-1: Intercellular adhesion molecule 1; IL: Interleukin; kdrl: kinase insert domain receptor; MMP: Matrix metalloproteinase; NET: Neutrophil extracellular trap; RANTES: Regulated on activation, normal T cell expressed and secreted; TAM: Tumor-associated macrophage; TGF-β: Transforming growth factor beta; TIE2: TEK tyrosine kinase; TSP-1: Thrombospondin 1; VCAM-1: Vascular cell adhesion molecule 1; VEGF: vascular endothelial growth factor; VEGFR: Vascular endothelial growth factor receptor.

TAMs and neutrophils are key drivers of metastasis, orchestrating processes like angiogenesis, immune evasion, and tumor cell migration. Their influence on the tumor–TME crosstalk and their extensive study in zebrafish models place them as central topics in this discussion. However, it is also important to acknowledge the contributions of other immune cells, such as NK cells, and B and T lymphocytes, whose complex interactions with the TME, while significant, fall outside the scope of this review.

## 3. Integration with AI and Computational Models

The increasing complexity of the biological data generated in zebrafish studies has driven the adoption of artificial intelligence (AI) and computational tools to analyze and interpret findings [146]. The ability to integrate advanced imaging techniques with AI-driven analysis has the potential to transform the zebrafish model into a data-rich platform for investigating metastasis and other diseases [146].

GBM zebrafish models have been significantly refined by applying advanced imaging techniques and computational analysis, providing a robust platform to study tumor progression and invasion [147]. Integrating high-throughput imaging with deep convolutional neural networks (CNNs) enables the precise tracking of tumor growth, vascular interactions, and invasive behaviors, including perivascular invasion and single-cell dissemination [147,148]. This combination allows for single-cell resolution imaging, reducing labor-intensive manual monitoring and enabling a more detailed understanding of how GBM cells interact with and adapt to their microenvironment [147]. Importantly, these models have also been used to evaluate patient-specific responses to therapies, such as the proteasome inhibitor marizomib, demonstrating consistent findings with other preclinical models [147].

In another example, the transparent *casper* zebrafish strain was used to develop a high-resolution imaging system that is capable of tracking tumor progression at a single-cell resolution [149]. In this study, quantitative imaging algorithms and a metastasis scoring framework (m-score) were introduced, allowing for the detailed characterization of tumor spread and metastatic burden in vivo [149]. Building on this foundation, integrating machine learning with imaging tools allows for a more efficient analysis of timelapse data to identify the molecular markers and behavioral patterns linked to aggressive tumor phenotypes [150,151]. For example, computational models could refine the analysis of metastasis-initiating cell (MIC) frequencies, invasion pathways, and TME interactions observed in zebrafish xenografts [149]. This approach could support early cancer diagnosis and the identification of patient-specific therapeutic targets.

Simulating therapeutic outcomes adds another dimension to the utility of zebrafish in cancer research. Integrating zebrafish-derived data with AI-driven simulations allows for the prediction of drug combination effects and the optimization of therapeutic regimens [152,153]. Specifically, computational models based on zebrafish drug screening results could simulate drug delivery dynamics, improving the design and precision of targeted therapies [154].

In conclusion, combining advanced imaging with computational tools enhances the capacity to model complex biological processes in zebrafish models while also supporting the development of early diagnostic tools and personalized therapies.

## 4. Advanced Drug Screening Pipelines and Translational Applications

Zebrafish models are increasingly essential for studying and testing anti-metastatic therapies, offering unique opportunities to integrate advanced methods such as PDXs and nanoparticle delivery platforms. These innovations enhance drug discovery and serve as a vital bridge between preclinical research and clinical applications.

### 4.1. Patient-Derived Xenografts in Personalized Medicine

Zebrafish patient-derived xenografts (zPDXs) provide an efficient platform for testing cancer behavior and drug responses in vivo in the context of personalized medicine (Figure 1B and Figure 2A) [35,155,156]. By transplanting tumor cells from individual patients into zebrafish embryos, researchers can rapidly evaluate therapeutic regimes, often delivering actionable results within days of sample collection [157]. zPDXs demonstrated a strong alignment with clinical outcomes, with studies indicating up to ~90% accuracy in predicting responses to chemotherapy and targeted therapies [155]. In this context, brain metastasis (BM) models revealed that microRNA-330-3p (miR-330-3p) enhances the invasive and colonizing capabilities of NSCLC cells. The overexpression of miR-330-3p increases the brain metastatic potential, while its knockdown suppresses metastasis, identifying it as a promising therapeutic target [26]. These models also revealed that although a mature blood–brain barrier (BBB) reduces metastatic efficiency, it does not entirely prevent cancer cell invasion. Live imaging highlighted the influence of the BBB on metastasis and provided a platform for evaluating therapeutic interventions. Notably, osimertinib, with its superior BBB permeability, outperformed gefitinib in treating BM [26].

The incorporation of advanced imaging techniques, such as selective plane illumination microscopy (SPIM) [158], further enhances the utility of zPDXs. In leukemia models, SPIM allowed for the assessment of distinct behaviors among cancer types, such as the rapid migration of leukemic cells within blood vessels [20]. In contrast to leukemia cells, breast cancer cells invaded the avascular tissues through a slower and amoeboid mode of migration [20]. These differences demonstrated the ability of zPDXs to capture the unique characteristics of various malignancies. Beyond investigating the metastatic mechanisms, this zebrafish model also demonstrated effectiveness in drug screening. For example, the rho-associated, coiled-coil-containing protein kinase 1 (ROCK1) inhibitor fasudil significantly reduced leukemic cell migration and overall survival [120], with quantitative metrics, like migration speed and dissemination distance, enhancing the evaluation of therapeutic responses [20].

zPDXs serve as an indispensable resource for advancing metastasis research and precision oncology due to their potential to rapidly model patient-specific tumor biology, predict therapeutic responses with a high accuracy, and uncover novel targets and mechanisms, ultimately contributing to developing more effective and personalized treatment strategies.

### 4.2. Nanoparticle-Based Drug Delivery

The transparent and vascularized nature of zebrafish embryos provides a unique platform for studying nanoparticle-based drug delivery systems (Figure 1G) [154,159]. This model allows for the real-time tracking of fluorescently labeled nanoparticles, enabling the detailed evaluation of their biodistribution, tumor uptake, and interactions with the metastatic microenvironment [160].

Light-triggered drug delivery systems are highly suitable for assessment in zebrafish models due to their embryos’ natural transparency [161]. Folate-modified liposomes, designed to enhance uptake via receptor-mediated endocytosis, have emerged as a promising approach for targeted drug delivery to tumors [162]. In zebrafish models, these liposomes, further stabilized with PEGylation to prolong the circulation time, achieved significantly greater tumor reduction compared with free-circulating drugs [162], demonstrating their potential in precise therapeutic strategies. In another study, zebrafish models were employed to evaluate G23-functionalized, curcumin-loaded zein nanoparticles (CUR-ZpD-G23 NPs), designed to cross the BBB and target GBM cells. This approach showed improved drug delivery with a higher cellular uptake and enhanced therapeutic effects, including increased apoptosis, elevated reactive oxygen species (ROS) production, and reduced cell migration and tumor growth [163].

Based on the previous applications, zebrafish show great potential for testing nanoparticles targeting the molecular pathways involved in metastasis, such as integrins and ECM components [154,160]. Arginine–glycine–aspartate (RGD)-functionalized nanoparticles, targeting integrins that are overexpressed on the tumor vasculature, effectively accumulated in breast cancer models both in vitro and in vivo in mice [164]. By disrupting the integrin-mediated pathways, these nanoparticles inhibited tumor adhesion and invasion, which are crucial steps in metastasis [164]. However, their application in other cancer types remains underexplored, and zebrafish could provide valuable insights into the broader efficacy of these systems and their ability to inhibit metastasis.

Together, these findings highlight the versatility of zebrafish as a model for nanoparticle-based cancer research, providing unique opportunities to investigate drug delivery, therapeutic responses, and molecular mechanisms across a range of tumor types and metastatic pathways.

### 4.3. Preclinical and Translational Importance

Zebrafish models provide substantial quantitative and statistical advantages in preclinical research. Their rapid development and high fecundity allow for the screening of up to 100 compounds per week, far surpassing the 10–20 compounds typically tested in murine systems [165]. This high throughput, combined with the robust statistical power from large sample sizes, enhances the reliability and reproducibility of the experimental findings [165]. Additionally, meta-analyses highlight the reliability of zebrafish models, demonstrating a predictive accuracy of approximately 80% in correlating drug efficacy with mammalian outcomes [166].

Combining high-throughput screening with mechanistic studies, zebrafish offer a cost-effective, efficient platform for preclinical research, reducing expenses compared with traditional mammalian systems and supporting early-stage drug discovery [24,165,166]. Despite their lower costs, zebrafish models maintain a strong predictive power for downstream mammalian validation, ensuring that results are both reliable and applicable to more complex systems [155,166].

Zebrafish embryonic processes, such as epiboly, share mechanistic parallels with cancer cell migration during metastasis [167,168]. Small-molecule inhibitors delaying epiboly in zebrafish, like the serotonin receptor 2C inhibitor pizotifen, also inhibit metastatic behaviors in human cancer cells [104]. Pizotifen’s anti-metastatic effects were confirmed in zebrafish xenografts and validated in murine models, highlighting zebrafish as a powerful tool for identifying effective therapies [104].

Along the same vein, zebrafish have proven effective in replicating the metastatic potential of various human cancer cells, including those from breast, prostate, colon, and pancreatic cancers [99]. For poorly invasive cell lines, zebrafish consistently show minimal metastasis, mirroring their in vitro behavior and validating their predictive accuracy [99]. Genetic studies in zebrafish, such as the knockdown of metastasis-promoting genes like *Wiskott–Aldrich syndrome protein family member 3* (*WASF3*) [169] or oncogenic kinases like Janus kinase (JAK)1 and JAK2 [46], demonstrated a significant reduction in metastatic spread. These results are consistent with findings from murine models, reinforcing the utility of zebrafish in metastasis research [170,171].

Moreover, zebrafish models are particularly well-suited for testing combination therapies, enabling the efficient evaluation of interactions between multiple agents [172]. Their unique capabilities not only streamline drug development but also refine hypotheses, validate therapeutic targets, and advance personalized treatment strategies. As a key component of preclinical and translational cancer research, zebrafish enhance the relevance and applicability of experimental findings, reinforcing their significance in the field.

### 4.4. Applications in Rare Cancers

Rare cancers with a high propensity for metastasis pose significant challenges in treatment development due to their low incidence, limited research focus, and insufficient funding [173]. Understanding their metastatic behavior and identifying therapeutic targets is crucial for developing effective, tailored interventions.

#### 4.4.1. Sarcomas

Despite advances in understanding Ewing sarcoma, the role of its hallmark *Ewing sarcoma breakpoint region 1-Friend leukemia integration 1* (*EWS-FLI1*) fusion gene in tumor progression and metastasis remains incompletely understood [32]. A zebrafish model of the disease provided a precise tool for addressing this gap, faithfully recapitulating its small-round-blue-cell morphology and aggressive gene expression profiles [174]. This model enabled the detailed analysis of *EWS-FLI1*, revealing its disruption of signaling pathways and the activation of oncogenic drivers like *NK2 homeobox 2* (*NKX2-2*) and *myelocytomatosis oncogene* (*MYC*), which are critical for neural development, tumor growth, and invasion [32,174].

Building on these insights, zebrafish models have proven valuable for high-throughput drug screening, enabling the identification of compounds that target *EWS-FLI1* or its downstream signaling pathways, such as MAPK and PI3K/AKT, which are critical for tumor cell survival and dissemination [175]. This highlights the potential therapeutic strategies to inhibit pathways driving tumor invasiveness and metastatic adaptation, advancing efforts to translate molecular findings into effective treatments.

#### 4.4.2. Pediatric Cancers

##### Neuroblastoma

Zebrafish models of neuroblastoma uncovered the impact of co-expressing the *MYCN* oncogene and *LIM domain only 1* (*LMO1*), which drives aggressive tumor growth and metastasis resembling high-risk human neuroblastoma [31,176]. Tumors in these models mirrored their human counterparts morphologically and exhibited increased ECM stiffness and collagen deposition, which are key factors enhancing metastatic potential [176,177].

##### Medulloblastoma

Medulloblastoma (MB) is another rare cancer that benefits from the use of zebrafish models. Zebrafish enable the orthotopic engraftment of MB cells, with the tumors developing in the hindbrain, closely replicating the localization of MB in humans [178]. Pre-culturing MB cells in a neural stem cell-like medium enhances their ability to home to the hindbrain, mimicking the neurotropism observed in human MB [178]. This process upregulates genes such as *Semaphorin 3A* (*SEMA3A*) and *Ephrin B1* (*EFNB1*) and induces a migratory neuronal phenotype associated with poor survival [178]. In another approach, transcription activator-like effector nuclease (TALEN)-mediated inactivation of tumor suppressor genes, such as *cyclin-dependent kinase inhibitor 2A* (*cdkn2a/b*) and *retinoblastoma 1* (*rb1)*, in zebrafish resulted in the development of tumors that mimic human MB and primitive neuroectodermal tumors (PNETs) [179]. RNA sequencing revealed the activation of pathways involved in cell cycle regulation, DNA replication, and neuronal development, including wingless-related integration site (WNT), sonic hedgehog (SHH), and NOTCH signaling, which are associated with distinct MB subtypes [179]. These pathways not only drive tumor growth but also regulate key processes such as migration and invasion [178,179].

CRISPR-based models have been developed to study SHH MB by knocking out *patched 1* (*ptch1*), a key genetic driver of this cancer [180]. Tumors arising from *ptch1* loss closely resemble human SHH MB in both histological and genomic features [180]. Combining the *ptch1* knockout with a *tp53* mutation resulted in more aggressive tumors, reflecting the poorer outcomes and increased risk of dissemination seen in certain human SHH MB cases [180,181]. Molecular players such as G-protein-coupled receptor (GRK)2/3 kinases were identified as key regulators of both tumor growth and behaviors related to migration and invasion [180]. The inhibition of GRK2/3 significantly slowed tumor progression and decreased markers of invasive capacity, highlighting their role in enabling cancer cells to breach local barriers and potentially metastasize to distant sites [180].

These findings demonstrate how zebrafish models provide critical insights into MB biology, supporting the development of novel therapeutic strategies to target both tumor growth and the mechanisms driving invasion and metastasis.

## 5. Addressing the Limitations of Zebrafish Models in Metastasis Research

As highlighted in this review, zebrafish have become a crucial model in cancer research, offering unique advantages such as optical transparency for real-time imaging and the potential for high-throughput studies [34,55,113,182]. These attributes make them invaluable for studying tumor progression and metastasis. However, limitations in replicating human disease complexity, such as differences in immune function [183], stromal composition [184], and TME [184], highlight the need for further research. Innovations like humanized zebrafish [185] are essential to overcoming these challenges and unlocking their full preclinical and translational efficacy.

There are several differences between the immune systems of zebrafish and humans [183], which can create challenges when studying immune-related metastatic processes. Larval zebrafish, which are the most suitable for imaging studies, lack a mature adaptive immune system [186]. Although this difference posits an advantage when generating xenografts, it can also hinder studies aimed at the adaptive immune system. Additionally, while adult zebrafish possess an adaptive immune system, it differs from that of humans in several key aspects [183]. For example, zebrafish produce immunoglobulins such as IgZ/T, which are absent in humans [187], and cannot produce IL-18 [188]. To overcome these issues, several zebrafish models that express human immune components, including cytokines like CXCL12 and granulocyte-macrophage colony-stimulating factor (GM-CSF) have been created [185]. These humanized zebrafish have been used to study cancer cell migration, metastasis, and the interaction of cancer cells with immune cells [185]. Additionally, the transplantation of human adaptive immune cells, such as T lymphocytes, has been employed to compensate for the absence of a fully developed adaptive immune system in zebrafish, enabling a more comprehensive exploration of tumor–immune dynamics [189].

The TME plays a critical role in metastasis, but differences between the zebrafish and human components pose challenges for modeling certain interactions. Zebrafish TME differs from its human counterpart, particularly in cellular composition and regenerative capacity [184,190]. Zebrafish stromal cells retain a progenitor-like phenotype that is different to human and mouse fetal stromal cells, which is associated with a robust ability to support tissue repair and regeneration [190,191,192]. While this feature enables dynamic remodeling of the ECM and robust cell–cell signaling, it does not fully mimic the characteristics of adult human stromal cells, which are more differentiated and less regenerative [184]. The zebrafish ECM also differs significantly in stiffness and molecular composition [98], which can influence key processes like tumor migration and invasion. Zebrafish ECM is often less rigid than that of human tissues, a factor that can affect mechanical movement and the behavior of invading cancer cells [98]. Advances in single-cell transcriptomics and cross-species analyses have highlighted these differences, showing that zebrafish stromal cells have a transcriptional profile that is more aligned with regeneration than with the tumor-promoting roles observed in adult human stromal cells [184]. Despite these differences, as previously described in this review, zebrafish models have been crucial in outlining the key molecular drivers of metastasis within the TME (Table 1). These include integrins, which mediate tumor–ECM adhesion and can be blocked to reduce cancer spread [193,194]; VEGF, which is essential for angiogenesis and studied using zebrafish with fluorescently labeled blood vessels [70,110,195]; TGF-β, which influences EMT and immune evasion [105,196]; EVs, which help to establish pre-metastatic niches [101,197]; and MMPs, which enable tumor invasion by degrading the ECM [50,198].

In zPDXs, tumors are engrafted along with the components of their original TME, including stromal, immune, endothelial cells, and microbiome [34]. Introducing foreign TME components into a zebrafish host presents challenges, as interactions between the host and donor TME can alter cell behavior, disrupt signaling pathways, and modulate immune responses [199]. For example, the zebrafish immune system may react to human-derived cells, potentially triggering inflammatory or rejection responses that could confound the experimental results [200]. Despite this, the inclusion of the donor TME can also provide significant benefits. Retaining elements of the original TME can allow for a more accurate replication of the tumor’s native microenvironment [201,202]. This is particularly valuable for studying tumor–TME dynamics, immune evasion mechanisms, and the effects of therapeutic agents in a more human-like context [201]. Moreover, the combination of the donor TME with the zebrafish host environment offers a unique opportunity to explore cross-species interactions. A recent study demonstrated that zebrafish-derived tumor necrosis factor-alpha (TNF-α) can directly induce apoptosis in human cancer cells by engaging their receptors, with macrophages polarized to a pro-inflammatory state driving this effect [203]. This finding highlights that human cells are not only responsive to zebrafish-secreted factors, such as cytokines, but also that they respond through conserved pathways, mimicking the behaviors observed in human physiological settings [203,204]. Additionally, the behavior of the donor TME in a foreign host can help to understand how environmental cues influence tumor progression [36]. By carefully optimizing conditions, such as inducing immune suppression in the host, matching ECM characteristics, or utilizing transgenic zebrafish lines expressing human proteins, it is possible to maximize the advantages of zPDXs. These approaches enable the study of complex tumor–host dynamics while preserving the translational relevance of the original TME.

Zebrafish models have proven to be an invaluable platform for uncovering key aspects of tumor metastasis and progression, despite their inherent limitations in fully replicating human biology. With innovations like humanized zebrafish and methods to incorporate human TME components, these models are evolving to address the current challenges, offering unique opportunities to study tumor–host interactions.

**Figure 2 ijms-26-00871-f002:**
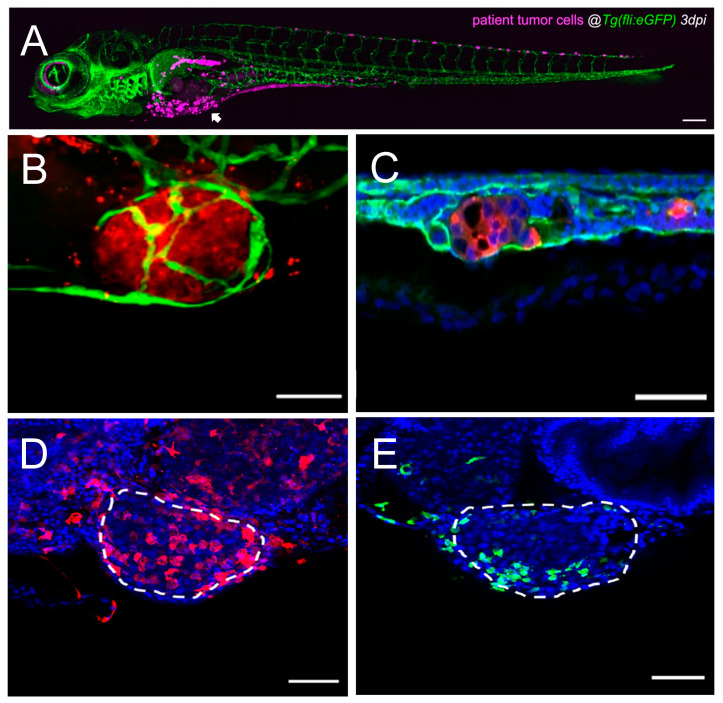
**Single-cell high-resolution visualization of the tumor–TME dynamics in the zebrafish model.** (**A**) Representative confocal image of an overview of a CRC patient-derived zebrafish xenograft 3 dpi, where human cells are labeled in magenta, and blood and lymphatic vessels are labeled in green. The arrowhead points towards the primary tumor, and human cells (magenta) are also observed in metastatic sites. Scale bar 200 µm. Adapted from ref. [155]. (**B**) Representative confocal image of a primary tumor in a CRC zebrafish xenograft 4 dpi, where human cells are labeled in red, and blood and lymphatic vessels are labeled in green. Scale bar 50 µm. Adapted from ref. [110]. (**C**) Representative confocal image of a metastatic tumor in the CHT of a CRC zebrafish xenograft 4 dpi, where human cells are labeled in red, blood and lymphatic vessels are labeled in green, and nuclei with DAPI counterstaining in blue. Scale bar 50 µm. Adapted from ref. [110]. (**D**) Representative confocal image of a primary tumor in a bladder cancer zebrafish xenograft 4 dpi, where macrophages are labeled in red, and nuclei with DAPI counterstaining in blue. White dashes outline the tumor. Scale bar 50 µm. Adapted from ref. [203]. (**E**) Representative confocal image of a primary tumor in a bladder cancer zebrafish xenograft 4 dpi, where neutrophils are labeled in green, and nuclei with DAPI counterstaining in blue. White dashes outline the tumor. Scale bar 50 µm. Adapted from ref. [203]. CRC: colorectal cancer; CHT: caudal–hematopoietic tissue; dpi: days post injection; TME: tumor microenvironment.

## 6. Future Directions and the Evolving Role of Zebrafish in Research

This review has summarized the key findings demonstrating how zebrafish models contribute to comprehending the complex mechanisms through which tumors evade their primary sites and home distant tissues. Current work is now focusing on improving the biological accuracy and experimental flexibility of zebrafish models to enhance their value in translational research [205,206,207,208]. One promising approach involves zebrafish models expressing human-specific molecules and components, including stromal elements [185,209,210,211,212]. This emerging strategy has shown potential [185] but requires further development to better replicate the complex TME and produce findings that are more relevant to human biology.

Combining patient-derived tumor organoids (PDTOs) with zebrafish models presents an unexplored opportunity to study tumor–TME interactions in a more dynamic and physiologically relevant context [213]. PDTOs capture the three-dimensional architecture of tumors, offering insights into individual patient biology, but lack the systemic dynamics of a living organism [213]. On the other hand, zPDXs allow for the study of these organism-level processes, though certain tumor characteristics may make direct engraftment challenging [34]. By transplanting PDTOs into zebrafish, it may be possible to integrate the structural fidelity of organoids with the systemic dynamism of zebrafish, overcoming limitations in both models. This approach could provide a versatile platform for investigating cancer progression, including metastasis and immune interactions, while maximizing resources through the high-throughput and cost-effective nature of zebrafish.

Advances in imaging technologies and genetic tools have further expanded the utility of zebrafish models in cancer research [54,55,114,175,182]. Fully optically transparent zebrafish strains now allow for the real-time observation of metastatic behavior over extended periods, even in adult zebrafish, enabling the detailed exploration of the spatial and temporal dynamics of tumor dissemination [214]. Meanwhile, the refinement of gene-editing tools like CRISPR-Cas9 has facilitated the creation of zebrafish models with precise genetic mutations linked to metastasis [25]. Combining these advanced imaging and genetic approaches opens new avenues to directly correlate tumor behaviors with their underlying genetic drivers, enhancing our understanding of cancer progression.

Beyond cancer research, zebrafish models have made significant contributions to broader areas of medicine and biology. Studies of cell migration in zebrafish have elucidated mechanisms of tissue invasion, which are applicable to both cancerous and non-cancerous conditions [215,216,217]. Research on ECM remodeling, a key factor in metastasis, has provided valuable insights into wound healing and tissue regeneration, advancing regenerative medicine [27,98,192]. Zebrafish studies on inflammatory mediators have the potential to contribute to the development of immunotherapies for cancer and autoimmune diseases [124,189,203,218,219]. Together, these findings demonstrate the versatility of zebrafish as a model system, offering unique opportunities to explore complex biological processes that extend beyond oncology.

In addition to their technical advantages, zebrafish offer exceptional accessibility and affordability, making them particularly valuable in resource-limited settings [34]. Their low maintenance costs enable high-throughput screening and advanced research, providing a cost-effective alternative to mammalian models, even in emerging laboratories [165]. This accessibility allows researchers in developing countries to model tumor dynamics relevant to their specific geographic and clinical realities. By focusing on the cancers that are most prevalent in their populations, these studies yield findings that are directly applicable to local healthcare challenges while contributing to global cancer research efforts. This approach fosters inclusivity and collaboration, driving impactful discoveries and advancing cancer diagnosis and treatment worldwide.

## 7. Conclusions

Zebrafish models are invaluable for studying cancer metastasis, providing key insights into the molecular drivers of tumor progression, angiogenesis, and tumor–TME interactions. Their optical transparency, rapid development, and genetic tractability allow for the real-time analysis of processes like ECM remodeling and vascular invasion. Advances in gene editing, imaging, and PDX technologies have further enhanced their role in exploring metastatic mechanisms and testing therapies. Accessible and cost-effective, zebrafish empower researchers worldwide, including those in resource-limited settings, to advance metastasis research. Despite some limitations, innovations such as humanized zebrafish are addressing these challenges, establishing zebrafish as critical tools for deepening our understanding of metastasis and advancing cancer research.

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
