# Peer review of "Small Fish, Big Answers: Zebrafish and the Molecular Drivers of Metastasis"

_ijms, 2025, doi:10.3390/ijms26030871_

Round 1
Reviewer 1 Report
Comments and Suggestions for Authors
In this manuscript, Mayra Fernanda Martínez-López presented the contribution of the zebrafish model to the study of metastatic cancers, taking advantage of the zebrafish's small size, optical transparency, genome-editing tools and fluorescent transgenic lines to carry out in vivo investigations. The review described how this biomedical model emerged for studying tumour-TME interactions at the cellular level using advanced imaging methods: extracellular matrix remodelling, angiogenesis, immune cell dynamics and tumour dissemination. It also highlighted the use of zebrafish genomic studies and high-throughput drug screening in preclinical studies at the interface between patient-derived tumour organoids and mammalian models of patient-derived xenografts, being therefore considered as a valuable model for personalised medicine. At the end, the review also addressed some of the limitations of the zebrafish and perspectives in cancer research.
The manuscript is well written but the different parts could be re-organized to facilitate the reading.
In the introduction, the presentation of the zebrafish is too succinct (lines 47-58) and does not address the major points that constitute the model's major advantages for cancer research: small vertebrates with conserved organogenesis, metabolism and physiology, available genome editing tools, conservation of the majority of cancer-associated human genes both in structure and function and signaling pathways (cell proliferation, migration, death, differentiation and immunity: DAMP signaling), permissive to human cell engraftment.
It needs to be completed to address more specifically the advantages of the model exploited in cancer research as presented later in the manuscript on lines 263 to 269.
A global presentation of the metastatic cancer models described in zebrafish should be added in the introduction.
In addition, pictures of fluorescence imaging should be included in the manuscript for example to illustrate the study of tumor angiogenesis in Zebrafish fluorescent transgenic lines (lines 156-159).
The second part Tumor-TME crosstalk: a driving force in metastasis is devoted to a description of molecular and cell biology studies that could be instead fused to the third part to improve the manuscript.
This will instead present advances in cancer research on a different perspective focusing on zebrafish model and emerging technologies instead of basic science. This can be done including the subparts of part 2 in part 3: the loss of functions (move paragraph 2.1 on MMP proteins and paragraph 2.2.1 on VEGF pathway, which can also be used to complete the part on drug screening), gain of functions, advanced imaging of cell-cell interactions or cell dynamics (move paragraph 2.2 on angiogenesis, paragraph 2.2.1 on Angiopoietin and ECM remoding, ...) and modeling of drug screening (move lines 175-190 ,.....). In this mode of presentation, a paragraph should be included relative to immune system in cancer to include the part 2.3.
Author Response
In this manuscript, Mayra Fernanda Martínez-López presented the contribution of the zebrafish model to the study of metastatic cancers, taking advantage of the zebrafish's small size, optical transparency, genome-editing tools and fluorescent transgenic lines to carry out in vivo investigations. The review described how this biomedical model emerged for studying tumour-TME interactions at the cellular level using advanced imaging methods: extracellular matrix remodelling, angiogenesis, immune cell dynamics and tumour dissemination. It also highlighted the use of zebrafish genomic studies and high-throughput drug screening in preclinical studies at the interface between patient-derived tumour organoids and mammalian models of patient-derived xenografts, being therefore considered as a valuable model for personalised medicine. At the end, the review also addressed some of the limitations of the zebrafish and perspectives in cancer research.
The manuscript is well written but the different parts could be re-organized to facilitate the reading.
Thank you to the reviewer for their valuable comments and suggestions, which have significantly enhanced the quality of the manuscript.
In the introduction, the presentation of the zebrafish is too succinct (lines 47-58) and does not address the major points that constitute the model's major advantages for cancer research: small vertebrates with conserved organogenesis, metabolism and physiology, available genome editing tools, conservation of the majority of It needs to be completed to address more specifically the advantages of the model exploited in cancer research as presented later in the manuscript on lines 263 to 269.
A global presentation of the metastatic cancer models described in zebrafish should be added in the introduction.
Thank you for these observations. We have included a more detailed description of the zebrafish as a model for cancer and metastasis research in the introduction part of the manuscript.
In addition, pictures of fluorescence imaging should be included in the manuscript for example to illustrate the study of tumor angiogenesis in Zebrafish fluorescent transgenic lines (lines 156-159).
A new figure illustrating examples of microscopy images has been added.
The second part Tumor-TME crosstalk: a driving force in metastasis is devoted to a description of molecular and cell biology studies that could be instead fused to the third part to improve the manuscript.
This will instead present advances in cancer research on a different perspective focusing on zebrafish model and emerging technologies instead of basic science. This can be done including the subparts of part 2 in part 3: the loss of functions (move paragraph 2.1 on MMP proteins and paragraph 2.2.1 on VEGF pathway, which can also be used to complete the part on drug screening), gain of functions, advanced imaging of cell-cell interactions or cell dynamics (move paragraph 2.2 on angiogenesis, paragraph 2.2.1 on Angiopoietin and ECM remoding, ...) and modeling of drug screening (move lines 175-190 ,.....). In this mode of presentation, a paragraph should be included relative to immune system in cancer to include the part 2.3.
Thank you for these suggestions, the manuscript has been rearranged accordingly.
Reviewer 2 Report
Comments and Suggestions for Authors
In the current article titled 'Small fish, big answers: Zebrafish and the molecular drivers of metastasis' by Mayra Fernanda et al, the authors have summarized the use of Zebrafish as an invaluable model for metastasis research, including tumor invasion, vascular remodeling, and immune evasion, and serving as a platform for drug testing and personalized medicine. Moreover, the authors have described how Zebrafish provide advantages such as patient-derived xenografts and innovative genetic tools in furthering advanced cancer research. While this article offers a lot of insightful information and summarizes up to date details on current understanding of using Zebrafish as a cancer research model, there are following comments that need to be addressed to further improve the quality of this article:
1. What are the reasons that other traditional species (mice, rat) are not used for studying metastasis? Describe briefly in the introduction.
2. For Section 2.2, what tumors have upregulated VEGF, hypoxia, angiopoietin, TIE, ECM and MMP levels? Are there inhibitors specific to these tumors in clinical trials. Please describe briefly in each of these sub-sections.
3. For Section 2, what are the similarities and dissimilarities in the findings that are reported in Zebrafish model to that observed in humans?
4. Lines 175-190: What are the side-effects and/or toxicities with using these drugs in the clinic? Describe briefly stating the importance of research in Zebrafish model to further develop targeted therapies with lesser side-effects.
5. While the article briefly states about immune system of Zebrafish, please describe how this model could be useful for studying immune cell and microenvironment in regulating tumor cell metastasis.
6. Consider including a few lines on how metabolism in tumor cells, tumor microenvironment and other organs could contribute to studying metastasis in Zebrafish model.
Author Response
In the current article titled 'Small fish, big answers: Zebrafish and the molecular drivers of metastasis' by Mayra Fernanda et al, the authors have summarized the use of Zebrafish as an invaluable model for metastasis research, including tumor invasion, vascular remodeling, and immune evasion, and serving as a platform for drug testing and personalized medicine. Moreover, the authors have described how Zebrafish provide advantages such as patient-derived xenografts and innovative genetic tools in furthering advanced cancer research. While this article offers a lot of insightful information and summarizes up to date details on current understanding of using Zebrafish as a cancer research model, there are following comments that need to be addressed to further improve the quality of this article:
Thank you to the reviewer for their valuable comments and suggestions, which have significantly enhanced the quality of the manuscript.
- What are the reasons that other traditional species (mice, rat) are not used for studying metastasis? Describe briefly in the introduction.
Thank you for the comment. While traditional species like mice are commonly used to study metastasis, the original manuscript highlights the zebrafish as an alternative model to address the limitations of traditional models, primarily stemming from their anatomical features and high husbandry costs. These limitations have already been briefly outlined in the initial introductory paragraph (lines 36–45).
- For Section 2.2, what tumors have upregulated VEGF, hypoxia, angiopoietin, TIE, ECM and MMP levels? Are there inhibitors specific to these tumors in clinical trials. Please describe briefly in each of these sub-sections.
We have addressed the comment as follows:
- VEGF: A new paragraph describing its overexpression in both hematological and solid malignancies as well as its effects on cancer progression and resistance to therapy has been added.
- Hypoxia: An introduction about hypoxia as a hallmark of cancer has been included.
- Angiopoietin and TIE: A description of the cancers with elevated serum concentrations of ANGTP2 and its effect on prognosis has been introduced.
- ECM and MMPs: A list of cancers with elevated expression of MMPs is now included.
- For Section 2, what are the similarities and dissimilarities in the findings that are reported in Zebrafish model to that observed in humans?
Thank you for the observation. A new column highlighting key findings and differences in the metastatic process between zebrafish and humans is now included in Table 1.
- Lines 175-190: What are the side-effects and/or toxicities with using these drugs in the clinic? Describe briefly stating the importance of research in Zebrafish model to further develop targeted therapies with lesser side-effects.
Thank you for pointing this out. This has been addressed now in the new manuscript.
- While the article briefly states about immune system of Zebrafish, please describe how this model could be useful for studying immune cell and microenvironment in regulating tumor cell metastasis.
As outlined in the manuscript, while the metastatic cascade involves various subsets of the immune system, we have chosen to focus our discussion on the two most extensively characterized immune subsets in the zebrafish model: macrophages and neutrophils. Due to space constraints, this targeted approach allows for a detailed examination of their roles within the metastatic cascade, which is comprehensively addressed in the manuscript. This model's value in elucidating the contributions of these immune cells to metastasis is thoroughly demonstrated and supported throughout the text.
- Consider including a few lines on how metabolism in tumor cells, tumor microenvironment and other organs could contribute to studying metastasis in Zebrafish model.
Thank you for the suggestion, we have added information regarding metabolism and its effects on tumor cells, specifically in the section of Hypoxia.